# A Decade of Discovery—Eukaryotic Replisome Disassembly at Replication Termination

**DOI:** 10.3390/biology13040233

**Published:** 2024-03-31

**Authors:** Rebecca M. Jones, Alicja Reynolds-Winczura, Agnieszka Gambus

**Affiliations:** 1Institute of Cancer and Genomic Sciences, Birmingham Centre for Genome Biology, University of Birmingham, Birmingham B15 2TT, UK; rebeccajones09@gmail.com (R.M.J.); a.reynoldswinczura@bham.ac.uk (A.R.-W.); 2School of Biosciences, Aston University, Birmingham B4 7ET, UK

**Keywords:** replication termination, CMG helicase, replisome disassembly, MCM2-7, ubiquitylation

## Abstract

**Simple Summary:**

During cell division, DNA is duplicated through a process called DNA replication, so that each new cell inherits a copy of its own. A high level of accuracy is essential in this for the maintenance of genome stability and the prevention of genetic disorders and ageing-related diseases. In this review, we describe the current knowledge around DNA replication termination, in particular comparing and contrasting the proteins and mechanisms identified in different organisms—from archaea through to humans—but with a specific focus upon eukaryotic replication termination. We discuss when and where termination takes place, the mechanisms of replication fork convergence and the process of replisome disassembly, in both S-phase and in mitosis. Recent advances in the field have revealed high levels of regulation in the process of replisome disassembly, demonstrating the importance of timely and appropriate unloading of replication machinery. Finally, we summarise how replication termination defects may impact cellular health and raise questions to be addressed in the future within the field.

**Abstract:**

The eukaryotic replicative helicase (CMG complex) is assembled during DNA replication initiation in a highly regulated manner, which is described in depth by other manuscripts in this Issue. During DNA replication, the replicative helicase moves through the chromatin, unwinding DNA and facilitating nascent DNA synthesis by polymerases. Once the duplication of a replicon is complete, the CMG helicase and the remaining components of the replisome need to be removed from the chromatin. Research carried out over the last ten years has produced a breakthrough in our understanding, revealing that replication termination, and more specifically replisome disassembly, is indeed a highly regulated process. This review brings together our current understanding of these processes and highlights elements of the mechanism that are conserved or have undergone divergence throughout evolution. Finally, we discuss events beyond the classic termination of DNA replication in S-phase and go over the known mechanisms of replicative helicase removal from chromatin in these particular situations.

## 1. Where and When Does Replication Termination Take Place?

The process of DNA replication comprises three stages: initiation, elongation and termination. During replication initiation in eukaryotic cells, the MCM2-7 (minichromosome maintenance 2-7) complex is loaded onto the DNA at origins in the form of double hexamers in a process called “origin licensing” (Figure 1i). The MCM2-7 complex forms the inactive core of the replicative helicase, and during S-phase, some (about 10%) of the MCM2-7 complexes are activated through their interaction with CDC45 and the GINS complex (composed of four subunits: Sld5, Psf1-3) (Figure 1ii). Once the CMG complex (CDC45, MCM2-7, GINS) has formed, the active CMG helicase travels through the chromatin, unwinding the duplex DNA and creating DNA structures known as replication forks. This stage of replication is referred to as elongation and involves the opening of the DNA so that interacting polymerases can copy each strand (Figure 1iii and the side box). Finally, DNA replication termination occurs wherever and whenever two replication forks originating from neighbouring origins converge (Figure 1iii), resulting in the creation of two separate double-stranded DNA molecules (Figure 1iv–vi). Historically described as a passive event, termination is often associated with the end of S-phase, but in reality, it takes place at all stages throughout S-phase [1].

Starting with the simplest organisms, prokaryotes (bacteria and archaea) and viruses typically possess only one single, circular mini-chromosome. The majority of these initiate replication from just one origin (*Ori*) and replication termination takes place on the opposite side of the chromosome (Figure 2). In *E. coli*, following replication initiation at *OriC*, termination is controlled by multiple replication barriers (terminators) placed throughout the chromosome. The Tus–Ter barriers consist of high affinity Tus proteins binding to specific DNA sequences called Ter. Such barriers can be orientated in a permissive orientation, permitting the bypass of the DnaB replicative helicase, or in a non-permissive orientation, which blocks the progression of DnaB. Following initiation, each fork bypasses five ‘permissive’ Tus–Ter protein–DNA complexes until they reach the stopping region. From this point onwards, progression is halted as the forks encounter the ‘non-permissive’ Tus–Ter complexes. It is here where the two forks thus converge and termination takes place (reviewed in [2,3]). It has been shown that without this tight control there is an accumulation of genomic instability and re-replication [4]. This is most likely caused by collisions between replication and transcription machineries, as the majority of bacterial genes are encoded on the leading strand and are thus transcribed co-directionally with replication (reviewed in [5]). The fact that multiple rounds of replication can take place at the same time on a bacterial chromosome, with new initiation events taking place before the previous round has completed [3], could also explain why terminating at a specific site is necessary. In bacteria, the sites of replication termination are, therefore, as well defined as the initiation sites.

Early research into DNA replication in the eukaryotic setting initially made use of the SV40 virus system, which also consists of a circular duplex DNA molecule. This virus exploits the host’s replication machinery during the infection of mammalian cells, encoding only for the replicative helicase, Large T antigen (T-Ag). It acts, therefore, as a very useful model for studying eukaryotic DNA replication. Like in bacteria, termination in SV40 was found to occur on the opposite side of the mini-chromosome to the origin. Unlike bacteria, however, if the origin is relocated, then the termination site also shifts, so it is clearly not a fixed site as it is in bacteria [6]. Although this system provides many clues about eukaryotic DNA replication, T-Ag is not controlled by the same regulatory mechanisms as the CMG helicase, leading to significant differences in the mechanisms of initiation and termination between this and eukaryotic chromosomal DNA replication.

Eukaryotic systems have much larger genome sizes and an increased number of origins is therefore essential to ensure that the entire genome can be replicated within the timeframe of S-phase. Importantly, the timeframe for S-phase varies between cell type and between embryonic and somatic cells. Eukaryotic embryonic model systems, *D. rerio* (Zebrafish) embryos and the *Xenopus laevis* egg extract system for example, complete DNA replication in just 15–20 min, respectively, supporting the fast nature of developing early embryos [7,8,9]. In order for this to occur, these systems display an extreme level of stochasticity in origin firing [7]. In somatic cells and as embryonic development progresses, the length of S-phase is increased as the replication factors become limiting, checkpoint signalling is activated and the need for gene expression arises, meaning that cells need to coordinate both replication and transcription; these all cause the frequency of replication fork initiation to decrease [7,9,10,11]. Despite the changes in the length of S-phase between these systems, 2D gel electrophoresis and autoradiography techniques have revealed that replication termination occurs in a non-site-specific manner in mammalian and yeast cells as well as in *Xenopus* early embryos [12,13,14] (Figure 2). 

The lack of specificity in termination sites in eukaryotes is a consequence of several factors including the following: (i) the origins are not sequence-specific; (ii) the origins fire at various times throughout S-phase; and (iii) the replication forks move at different speeds throughout the genome and encounter barriers. As termination generally occurs mid-way between two neighbouring fired origins [1,15], the site from which they initially fire will thus affect where they terminate. Aside from *S. cerevisiae*, which has sequence-specific origins, origin firing in eukaryotes is non-sequence-specific. In fact, there is a huge range of stochasticity in this, meaning that no two cells within the same population have identical patterns of origin usage [1], which is believed to confer robustness and reliability for DNA replication [7]. While origin firing is stochastic within topological domains of the genome, i.e., 3D structural domains generated within chromatin, duplication of the genome in fact follows a strict replication timing programme, with large segments of the chromosomes firing early in S-phase and others firing late in S-phase [16]. These clusters of origins are referred to as Initiation Zones (IZs) and those firing in the early S-phase tend to correlate with DNase I hypersensitive sites and active histone marks [17], suggesting that these origins are selected on accessible, nucleosome-free regions of DNA [18]; such replicons are likely to terminate even before the origins in the late S-phase have fired.

As well as being influenced by the choice and timing of origin firing, sites of termination are also heavily influenced by the speeds at which the individual, converging forks are travelling. Though the majority of replication forks travel at an average speed of 1.5 kb/min [19], they will travel faster or slower in different parts of the genome, depending on the environment. Within the context of chromatin, replication forks may encounter many problems along the way, such as nucleosome displacement errors, difficult-to-replicate DNA sequences, poor availability of dNTPs, obstacles such as DNA–protein barriers or crosslinks (DPCs), collisions with transcription machinery and difficulties with resolving torsional stress (Figure 2). Indeed, many factors travel with the replication forks to aid their progression, such as histone chaperones, additional helicase enzymes and topoisomerases.

Endogenous barriers to replication also exist within eukaryotes, akin to those described within bacteria. For example, the replication fork barrier (RFB), which comprises the Fob1 protein complexed with the ribosomal DNA (rDNA) repeat within *S. cerevisiae*, ensures that unidirectional replication occurs throughout this region [20]. Also, the replication termination sequence 1 (RTS1) at the fission yeast mating-type locus ensures unidirectional replication. For an excellent review on these and other replication pause sites, see [21]. More recently, and relevant to this review, it has been proposed that inactive/dormant MCM2-7 double-hexamers themselves act as barriers to progressing replication forks in both yeast and in human cells [22,23]. The current rational is that they work to maintain a healthy speed of replication fork movement, but as a result, they are also likely to influence the site of termination.

More recently, high resolution sequencing techniques in both yeast and mammalian cells have confirmed that termination events/zones are more heterogeneous than initiation events [17,24] and tend to enrich with repressive histone marks and localise within intergenic regions [17]. Such termination zones in human cells range in size from 120 to 500 kb [25] and are thus understandably very difficult to isolate and analyse (Figure 2). Within an asynchronous population of DT40 chicken B cells for example (a model cell line used for its ease in performing genetic manipulation), only 8% of the population of replication forks analysed through the DNA fibre assay were converging at any one time.

## 2. Replication Fork Convergence

Once obstacles have been removed and the two helicases come into close proximity (~150 base pairs), the last stretch of parental DNA between the forks needs to be unwound, a process referred to as dissolution [26,27]. The accumulation of positive supercoils and torsional stress between the forks, however, means that unwinding this last stretch of DNA is a topologically challenging event, and so several active mechanisms have been identified which help to complete the process.

The first mechanism of relieving this torsional strain is through clockwise fork rotation. This was initially identified in early studies using the SV40 system [28,29,30] and is now considered to be the primary mechanism utilised, because topoisomerase enzymes have limited access to these very last stretches of DNA [31,32]. As the forks rotate, the torsional strain is passed behind them, in the form of double-stranded intertwines called pre-catenanes. Fork rotation is restricted specifically to termination and to particular pause sites only, by the actions of TIMELESS/TIPIN (Tof1/Csm3), so as to prevent the accumulation of DNA damage, particularly at fragile sites, thus protecting genomic stability [33].

Following fork rotation, the pre-catenanes generated behind the replication forks must now be processed by Topoisomerase II (TopII) enzymes [34,35]. Untangling the intertwined DNA helices not only ensures correct separation during mitotic division, so that each daughter cell receives an equal amount of DNA, but it also influences fork convergence and the completion of DNA synthesis. TopII enzymes perform this function by generating transient double-strand nicks in the backbone of one duplex, whilst threading the second duplex through the opening [36]. Such examples of this include TopIV in bacteria [37], TopII in SV40 [30], TopII in *S. cerevisiae* [38] and TopIIα in *Xenopus* egg extract [27]. 

In yeast and *Xenopus* systems, however, the depletion of TopII does not prevent fork convergence; rather, it just causes a delay [27,38]. This indicates that eukaryotic systems have evolved alternative or additional mechanisms to aid fork convergence, which are independent of topoisomerase. Such mechanisms are now being identified. Supporting earlier work from the Foiani group, Deegan et al. (2019) recently revealed that the 5′-3′ helicases Pif1 and Rrm3 support the full dissolution of parental DNA in the absence of TopII in *S. cerevisiae* [38,39], while Campos et al. (2023) also found that the 5′-3′ helicase RTEL1, along with MCM10, can promote this process in *Xenopus*, suggesting that such helicases either work redundantly to, or in concert with, topoisomerases [40]. Given that these enzymes have opposite polarity to the MCM2-7 helicase, it is thought that the 5′-3′ activities of these helicases on the lagging strand template complement the 3′-5′ unwinding activity of the MCM2-7 helicase on the leading strand template (Figure 1iv). 

Finally, although the global analyses mentioned above indicate that termination sites tend to localise to intergenic regions, Choudhary et al. (2023) found that, in yeast, Rrm3 is able to promote fork convergence at termination sites (TERs) which specifically reside within transcription units. In this case, Rrm3 coordinates with the helicase Sen1 to help counteract RNA Polymerase II and RNA-DNA hybrids, thus enabling the full completion of fork convergence and replication. In the absence of both factors, RNA Pol II accumulates at TERs and cells express chromosomal fragility [41].

## 3. Completion of DNA Synthesis

In 2015, Dewar et al. set out to establish the mechanism of vertebrate replication termination [26]. Using a highly synchronised plasmid-based assay in the *Xenopus* egg extract system, they demonstrated that once the two converging helicases had made contact, they were able to pass one another (Figure 1iv–vi). This was then further corroborated by their single-molecule studies in 2020 [42]. This striking observation came hand-in-hand with the change in thinking towards origin initiation: prior to 2017, it was generally accepted that progressing replication forks travelled with the C-terminal tier of the MCM2-7 helicase at the front of the replisome. Important structural studies performed by the O’Donnell and Diffley groups, however, revealed that the helicase in fact travels with the N-terminal tier at the front. This means that the helicases must pass one another at initiation before they move away in a bi-directional manner (Figure 1i–iii) [43,44], just as observed at termination.

During replication termination, and after passing one another, each helicase ultimately reaches the 5′ end of the previous Okazaki fragment. Dewar et al. (2015) proposed that the helicases then slide onto these ssDNA-dsDNA junctions and eventually encircle the dsDNA (Figure 1v–vi). This was verified by single molecule imaging, which revealed that terminated CMGs can continue translocating along DNA substrates but are incapable of supporting DNA synthesis, supportive of their positioning on dsDNA [42]. Indeed, early experiments investigating the helicase activity of the human CMG complex also showed that, if the helicase meets a ssDNA-dsDNA junction, it is able to slide onto the dsDNA duplex and translocate along it [45]. When thinking about this junction specifically at termination, however, it is important to remember that the helicase will in fact encounter an RNA-DNA hybrid, due to the presence of the 4–12 nt RNA primer of the last immature Okazaki fragment (Figure 1v, red arrow). Studies to understand replication–transcription collisions in bacteria have shown that the DnaB replicative helicase stalls at RNA-DNA hybrid structures (R-loops) on the leading strand template [46,47], although the shorter species posed less of an obstacle to helicase progression. It remains to be determined what effect, if any, this short RNA molecule has on the CMG helicase at termination and to find out how this RNA molecule is ultimately processed. Either way, we know that following this encounter, the helicase translocates onto the duplex dsDNA molecule and moves away from the termination site, as has been observed in several in vitro experiments [42,48,49]. 

With the CMG helicases having now completed their roles, this leaves the very last stretches of DNA to be synthesised and the RNA primers of the last Okazaki fragments to be removed. Interestingly, researchers working with the SV40 system found that replication forks stalled prior to termination [50] and that the two circular daughter molecules segregated before the completion of DNA synthesis [51]. This resulted in the creation of two daughter molecules containing persistent 22–73 nt long single-stranded gaps at their termination sites [51]. In contrast, Dewar et al. (2015) detected only a short 3 nt gap between the leading and lagging strands in the eukaryotic plasmid-based system, highlighting significant differences between these systems. They also observed rapid ligation between the leading strand and the last Okazaki fragment of the lagging strand, as visualised by denaturing agarose gel electrophoresis. This ligation was found to occur in the same timeframe as RPA’s (Replication Protein A) dissociation from the DNA. As RPA binds and protects ssDNA from breakage and degradation, this dissociation pattern suggests that RPA coats and protects sequences of ssDNA within the termination zone [52,53].

## 4. Replisome Disassembly

The final step in replication termination, which has arguably dominated the field, involves ubiquitylation and the disassembly of the terminated helicases from chromatin through the actions of Cullin ubiquitin ligases and the p97/Cdc48/VCP protein unfoldase (Figure 3). 

Although early observations made in the SV40 virus system suggested that the T-Ag helicase dissociates from the DNA prior to dissolution [54], Dewar et al. (2015) demonstrated that, in vertebrates, disassembly of the replisome from chromatin occurs after dissolution, ligation and decatenation. As discussed above, it is now widely accepted that disassembly of the CMG helicase takes place from the dsDNA [26]. This process has now been studied in several different organisms including *S. cerevisiae*, *X. laevis*, *C. elegans* embryos and, more recently, mammalian cell lines. The data gathered from all of these studies mean that we now have a great deal of understanding about this highly conserved mechanism.

First described in the yeast *S. cerevisiae* and in the vertebrate cell-free *Xenopus* egg extract system, polyubiquitylation of the MCM7 subunit of the replicative helicase is the central event in this process, determining whether CMGs will be unloaded from DNA or retained on chromatin [55,56] (Figure 3). While the initial studies revealed that MCM7 ubiquitylation at termination is dependent on the Cullin family of E3 ubiquitin ligases (discussed in depth below), further investigations showed that MCM7 is ubiquitylated specifically at a conserved lysine residue(s) within the N-terminal loop, which protrudes from the surface when complexed with MCM2-6. This residue was identified to be K29 in budding yeast Mcm7 [57], K27 and K28 in *Xenopus* Mcm7 [42] and K28 in human MCM7, in proteome-wide screens [58,59]. However, Maric et al. (2017) [57] found that additional lysines could be used in vivo when K29 was mutated in yeast, suggesting that there are redundant sites available. Indeed, we have also observed this for human MCM7 when it is mutated at K28 (unpublished data from the Gambus lab).

### 4.1. Cullin Ubiquitin Ligases Driving MCM7 Ubiquitylation

In *S. cerevisiae*, polyubiquitylation of Mcm7 was shown to be driven by the Cullin1-RING E3 ubiquitin ligase, known as SCF^Dia2^, along with the ubiquitin conjugating enzyme (E2) Cdc34 [60] (Figure 3i). The SCF^Dia2^ complex is an E3 ubiquitin ligase, consisting of the following four subunits: Cullin1/Cdc53, the scaffold protein, which provides a platform for binding of the other subunits; Rbx1 (Hrt1), a RING finger protein, which binds to the C-terminus of Cullin1; Skp1, the adaptor protein at the N-terminus of Cullin1 and finally, Dia2, the F-box substrate recognition subunit that recognises the target for ubiquitylation [61]. Dia2 has two protein–protein interaction motifs in addition to the F-box domain: a TPR domain at its N-terminal region and a large leucine-rich repeat (LRR) domain, which makes up a large part of the protein at the C-terminal end [62]. Unlike the TPR domain, the LRR domain, is indispensable for the function of Dia2, in counteracting replication stress and maintaining cell cycle function [63]. In the process of replisome disassembly, this particular region is now known to aid direct interactions between the SCF^Dia2^ and Mcm3,5,7 of the replicative helicase during termination [64,65]. The TPR domain, although not essential, has been found to promote interactions between Dia2 and components of the replication progression complex (RPC) in yeast, i.e., Mrc1 and Ctf4 [62]. This tethering of SCF^Dia2^ to the RPC, is now known to promote ubiquitylation of the CMG helicase upon termination as the loss of this tethering results in a partial CMG disassembly defect. Indeed, combining a disassembly-defective allele of Cdc48 with the loss of this TPR domain produces synthetic lethality in yeast [63].

As SCF^Dia2^ is exclusively a yeast protein, and is not present in metazoa, it was vital to identify the E3 ubiquitin ligase performing MCM7 ubiquitylation in higher organisms. Experiments in *C. elegans* and *X. laevis* thus identified a Cullin ligase CUL-2^LRR−1^/Cul2^Lrr1^ as a key protein recruited to chromatin at replication termination and confirmed that it was responsible for Mcm7 ubiquitylation [64,65]. More recent studies have now also confirmed CUL2^LRR1^ to be needed for replisome disassembly in mouse stem cells and human cell lines [66,67,68] (Figure 3i). These latter studies demonstrated that the loss or depletion of LRR1 by CRISPR knockout, auxin-inducible degradation or siRNA impaired the ubiquitylation of MCM7 and led to the accumulation of replisomes on chromatin.

CUL2^LRR1^ is a Cullin-RING E3 ubiquitin ligase, present only in metazoa. It consists of the following five subunits: CUL2 is the scaffold protein to which other essential ligase subunits bind; Rbx1 recruits the E2 ubiquitin conjugation enzyme; Elongins B and C act as adaptor proteins and partially resemble ubiquitin and the Skp1 protein, respectively [69]; and finally, Lrr1 is the substrate recognition receptor. In order for this complex to become active, CUL2 also possesses a neddylation site, which is modified by NEDD8 and is important for cullins’ ubiquitylation activity [70]. The exception to this is *S. cerevisiae,* in which neddylation is not essential for CMG ubiquitylation by Cullin1 as the deletion of the NEDD8 orthologue, Rub1, only modestly reduces CMG ubiquitylation [71].

MCM7 polyubiquitylation by CUL2^LRR1^ in higher eukaryotes seems to occur as a two-step process due to the cooperation of CUL2^LRR1^ with various E2 enzymes (Figure 3i). Using a reconstituted in vitro ubiquitylation reaction, *C. elegans* MCM7 was shown to be initially modified with a short ubiquitin moiety, by the actions of E2 ubiquitin ligases UBC-18_ARI-1 (hUBE2L3) and LET-70 (hUBE2D) [72]. These short chains are then extended by the E2 enzymes UBC-3 or UBC-7 (albeit less efficiently) to create long K48-linked polyubiquitin chains. Indeed, the same was observed for human in vitro reconstituted CUL2^LRR1^, which initially monoubiquitylates MCM7 within the context of the CMG, in cooperation with the UBE2D class of E2 enzymes. Following this, the UBE2R1/R2 or UBE2G1/G2 classes of E2s work to extend the chains, creating long, polyubiquitylated, K48-linked chains on MCM7 [73].

### 4.2. Recruitment of Ubiquitin Ligase to Terminated Helicases

In order to more thoroughly understand how these ubiquitin ligases are drawn to the terminated helicase, Jenkyn-Bedford et al. (2021) [74] applied cryo-electron microscopy (cryo-EM) analyses to visualise reconstituted post-termination budding yeast and human replisomes. These pivotal experiments using replisomes on dsDNA substrates or in solution enabled them to capture how ubiquitin ligases interact with the replisome. Despite significant architectural differences between the substrate receptors Dia2 and LRR1 (discussed further below), this work revealed that both SCF^Dia2^ and CUL2^LRR1^ interact directly within the N-terminal tier of the MCM2-7 helicase, across the zinc-finger domains of MCM3 and MCM5 (Figure 4). This was further confirmed through mutational studies of the residues seen to be engaging in this interaction; mutations in the LRR domain of Dia2, and in MCM3 and MCM5 (still proficient for DNA replication), produced a defect in MCM7 ubiquitylation, while yeast cells expressing this mutant Dia2 also displayed a replisome disassembly defect.

The LRR-domain-binding region is relatively far from the ubiquitylation site on MCM7 (Figure 4), so the question arose as to how the ubiquitin ligase is able to reach the target site. In answer to this, cryoEM structures of the yeast and human terminated replisomes in complex with the ubiquitin ligase revealed an elongated arm extending from the N-tier of the MCM2-7 helicase [74]. Despite the low resolution, indicative of a large degree of flexibility, the known crystal structures of yeast SCF-Skp1 and human ELOB-ELOC-CUL2-RBX1 could be unambiguously docked into these sites, showing that the catalytic modules of both SCF^Dia2^ and CUL2^LRR1^ are located 45–70 Å from the primary ubiquitylation site on Mcm7 once they have bound with the replisome. Notably, the authors speculated that the variable conformations in their structures, i.e., the flexibility of the E3 ligases, enables them to produce the elongating ubiquitin chains on MCM7. This proposed model is strongly supported by the detailed structures of the *X. laevis* Cul2^Lrr1^ complex, produced by the Brown and Walter labs; they co-expressed of all five subunits in the baculovirus system and cryo-EM analysis revealed that Cul2^Lrr1^ possesses a long, open architecture. This means that the LRR domain of the Lrr1 subunit is positioned far away from the catalytic Rbx module, again explaining how the LRR domain can interact with the MCM3,5 region of the replisome, while the Rbx module is able to reach the Mcm7 N-terminal loop for ubiquitylation [75]. 

Although these structural studies provide an enormous amount of knowledge about the mechanisms of E3 ligase recruitment to the terminating helicase, it is possible that certain factors could be missed due to the pre-determined nature of these reconstitution experiments and/or the dynamic nature of some interactions. Indeed, in vivo interaction studies, identifying replisome factors important for the recruitment of the E3 ligase, have revealed interesting differences between *S. cerevisiae* and *C. elegans*. In *S. cerevisiae*, the ligase activity of SCF^Dia2^ and polyubiquitylation of Mcm7 in the context of the CMG were stimulated by the presence of the RPC factor Mrc1 (CLASPIN in humans and CLSP-1 in *C. elegans*) and the trimeric Ctf4 (AND1 in humans and AND-1 in *C. elegans*). Without these factors, the ubiquitylation of Mcm7 was significantly impaired, with only short ubiquitin chains detected, and disassembly was defective [76]. Notably, these particular components of the replisome are positioned around Mcm2,6 of the helicase and around the Mcm2,5 gate, in proximity to Cdc45 and GINS, respectively (Figure 4). Combining this data with that discussed above, the current mechanism in budding yeast thus comprises SCF^Dia2^ interacting with Mrc1 and Ctf4 via the N-terminal TPR domain of Dia2, and with Mcm3,5,7 through the C-terminal LRR domain of Dia2; from here, Mcm7 can now be ubiquitylated at the N-terminal loop.

When investigating this mechanism within *C. elegans* early embryos, however, Xia et al. (2021) found that the recruitment of the CUL-2^LRR−1^ complex instead relies upon the RPC factors TIMELESS and TIPIN (TIM-1/TIPN-1 in *C. elegans* and Tof1/Csm3 in *S. cerevisiae*). Reconstituted reactions of ubiquitylation and replisome disassembly, as well as in vivo experiments, indicated that such factors help to recruit CUL-2^LRR−1^ to the CMG and also promote the priming and extension of ubiquitylation events on CMG-MCM7. Moreover, the recruitment of the CUL2 complex also involves the N-terminal pleckstrin homology (PH) domain within LRR1. Indeed, structural analyses revealed that this region interacts with the zinc-finger domains of MCM2,6, as well as with the N-terminal region of TIMELESS, further supporting the role of this particular RPC component in CUL2^LRR1^ recruitment [74] (Figure 4). Conversely, while CTF-4 and CLSP-1 had no detectable stimulating effect on MCM7 ubiquitylation [72], Jenkyn-Bedford et al. (2021) did in fact observe interactions between human CUL2^LRR1^ and the AND1 subunit in their structural analyses. Assuming that human and *C. elegans* CUL2^LRR1^ function in the same way, this suggests that, while such interactions do exist here, they are dispensable for recruitment and activity [74].

All of this research shows that, despite the clear differences in the regulation of recruitment mechanisms for SCF^Dia2^ in budding yeast and CUL2^LRR1^ in higher eukaryotes, which are most certainly brought about by the significant sequence and structural differences that reside in the substrate receptors, remarkably, the same patch of MCM3/5 ZnF domains are utilised for the main interaction and the same lysine residue(s) of MCM7 is targeted in all systems.

### 4.3. Restricting MCM7 Ubiquitylation to Termination

As research conducted in the different model systems resulted in the conclusion that ubiquitylation of CMG-MCM7 is a crucial event for effective replisome disassembly, the vital question arose: how is CMG-MCM7 ubiquitylation permitted and efficiently executed solely at replication termination, but blocked throughout the initiation and elongation phases of DNA replication? During replication fork progression, it is vital to maintain the structure of the MCM2-7 helicase because its loading is restricted to late mitosis/early G1 phase when CDK (Cyclin-Dependent Kinase) activity is low. This ensures once-per-cell-cycle replication and allows the avoidance of re-replication. As a result, re-loading of the helicase is not possible during S-phase and so premature unloading of helicases would lead to fork collapse and replication stress. One important role for checkpoint activation is thus to protect the helicase at stalled replication forks, enabling fork restart once the encountered damage has been resolved [77].

Given the research conducted so far, as discussed above, the primary mechanism by which this process is regulated is certainly through the selective binding of the E3 ligase to the CMG complex. We know that this is the first step in replisome disassembly and so recognition of and binding with the substrate are the first and foremost important regulatory mechanisms. Several theories were initially proposed to explain why and how the E3 ligase only binds with the CMG at termination, including: (1) CMGs encountering the previous Okazaki fragment undergo a conformational change and become more accessible for E3 ligase recruitment; (2) interactions between the two converging replisomes promote recruitment of the E3 ligase; (3) a protein or DNA structure at the elongating fork prevents recruitment of the E3 ligase, suppressing MCM7 ubiquitylation until termination. Over the years, research in both budding yeast and the *Xenopus* system has revealed the following facts: MCM7 in the context of the CMG can be ubiquitylated in the absence of any DNA or when the replicating DNA is eliminated through digestion with benzonase [42]; MCM7 ubiquitylation is prevented in the presence of a Y-shaped DNA substrate, i.e., the replication fork [76]; the helicase can be ubiquitylated and disassembled when encountering a single-strand nick within the lagging strand [78]. All of these observations pointed towards a mechanism by which lagging strand exclusion during replication elongation prevents MCM7 ubiquitylation at the active replisome.

Importantly, many structural studies had previously shown that the lagging strand passes through a channel between the zinc-finger domains of MCM3 and MCM5 [43,79,80,81] and, as discussed above, this is the precise, shared location at which both Dia2 and LRR1 interact [74]. Authors have speculated that this particular site is therefore obstructed until termination, preventing premature MCM7 ubiquitylation and MCMs disassembly. Prior to replication initiation, the site is likely to be obstructed by dimerisation of the two MCM2-7 hexamers at the origin. At initiation and elongation, the site is obstructed by the exclusion of the lagging strand. Then, at replication termination, when the helicase slides onto the dsDNA and is no longer bound with the forked DNA, the MCM3,5 interface finally becomes accessible to the E3 ligase, which performs MCM7 ubiquitylation.

### 4.4. p97/Cdc48/VCP

Ubiquitylation of the MCM7 subunit within CMG is the central event in replisome disassembly, but in fact, MCM7 only transiently remains in this modified form because it is rapidly recognised by the p97/Cdc48/VCP protein unfoldase and extracted from chromatin (Figure 3ii–iii). As such, it can be difficult to detect ubiquitylated MCM7 on chromatin, unless p97 activity is inhibited.

p97 complexes play important roles in the cellular environment: they unfold stable proteins and direct them for degradation by the proteasome, act on misfolded membrane proteins to extract them from the membranes for degradation, and dismantle large protein complexes through the unfolding of their subunits, e.g., the extraction of CMGs from the chromatin after the termination of replication (reviewed in [82]). p97 belongs to the AAA+ (ATPases associated with diverse cellular activities) family and forms a double-ring structure, which consists of six p97 monomers. Each monomer comprises an N-terminal tier and two ATPase domains, D1 and D2 [82,83,84]. Using ATP hydrolysis within the D2 domain, ubiquitylated substrates are translocated through the narrow central pore of the hexameric ring and are thereby unfolded [85]. Following this, they are either degraded by the proteasome or recycled.

#### 4.4.1. p97 Cofactors

In many cases, the activity of p97 relies upon cofactors/adaptor proteins, which commonly bind with the N-terminal tier of the complex and help to stimulate its processing activity. Given that p97 has such a plethora of substrates within the cell, it is thought that these cofactors also provide substrate specificity. Early work on the role of Cdc48 (p97 in *S. cerevisiae*) in replisome disassembly showed that its recruitment to ubiquitylated CMG requires the ubiquitin-binding, heterodimeric complex Ufd1/Npl4, which was identified in a screen for Cdc48 binding partners in budding yeast [57,86] (Figure 3ii). Ufd1/Npl4, along with p47, is the most well-studied cofactor for p97 and is known to aid in the processing of many different cellular substrates which have been modified with K48-linked polyubiquitin chains (reviewed in [87]). Demonstrating its specificity for ubiquitylated Mcm7 at termination, the binding of the budding yeast Cdc48^Ufd1/Npl4^ complex to CMG in vitro was shown to be dependent on Mcm7 K29 ubiquitylation [57]. In vitro reconstitution experiments, which defined the minimal requirements for budding yeast replisome disassembly, also confirmed that Ufd1/Npl4 cofactors promote this process [71]. 

Metazoa also use p97 protein for replisome disassembly; however, the functioning of p97 seems to be far more complex here than in yeast. In vitro and in vivo experiments conducted in different metazoan model systems revealed that, in addition to the UFD1/NPL4 major cofactor, full functional processivity of p97 also requires so-called minor cofactors from the UBA-UBX family: UBXN7, FAF1/UBXN-3 and FAF2 [72,88,89,90] (Figure 3ii). Notably, the mutation of these factors in yeast has no effect on CMG unloading [57].

Firstly, within *C. elegans*, CDC-48^UFD−1/NPL−4^ activity towards terminated CMG is stimulated by UBXN-3. Indeed, in the absence of TIMELESS-TIPIN, when CUL-2^LRR−1^ recruitment is compromised, UBXN-3 becomes essential for disassembly, as the co-depletion of UBXN-3 and TIMELESS causes synthetic lethality [64,72]. Examinations in mouse embryonic stem cells and in the *X. laevis* egg extract system on the other hand, revealed that two minor cofactors, UBXN7/Ubxn7 and FAF1/Faf1 (UBXN-3 homologue), both interact with p97^Ufd1/Npl4^ in S-phase but that UBXN7/Ubxn7 is the most crucial for timely CMG unloading [88,90]. Finally, in human cells, it seems that FAF1, FAF2 and UBXN7 all stimulate the activity of p97^UFD1/NPL4^ within reconstituted disassembly reactions, but FAF2 is unlikely to be involved in vivo as it is a membrane-anchored protein which is localised to the endoplasmic reticulum [88].

While there is controversy in the literature as to how stable the interactions between p97 and its minor cofactors are, the data that have been gathered suggest that, at least for Ubxn7, they stimulate efficient recruitment of p97^Ufd1/Npl4^ to the ubiquitylated substrate by bridging interactions between the ubiquitylated substrate, the Cullin ubiquitin ligase and p97 [89,90,91]. This bridging by Ubxn7 is made possible by its three domains: the UBA (ubiquitin-associated) domain, which interacts with the ubiquitin chain; the UIM (ubiquitin-interacting motif) domain, which interacts directly with the NEDD8 modification on Cullins [92] and the UBX (ubiquitin-regulatory X) domain, which binds with the amino-terminal N-domain of p97 [93]. 

Ubxn7 interacts with p97 in the cytoplasm of the egg extract and its recruitment to chromatin is dependent on the ability of p97^Ufd1/Npl4^ to bind with the ubiquitylated substrate [89,90]. This suggests that Ubxn7 and p97^Ufd1/Npl4^ are recruited to chromatin together. While Ubxn7 was found to be more important for timely CMG unloading than Faf1, it seems that unloading does eventually occur, even in the absence of Ubxn7. It was demonstrated that Cul2^Lrr1^ levels on chromatin are markedly increased in the absence of Ubxn7, with ubiquitin chains on Mcm7 becoming considerably longer. This lengthening of the chains likely stimulates the eventual binding of p97^Ufd1/Npl4^, as increasing ubiquitin chain lengths are known to drive p97 activity [90]. Alternatively, in the absence of Ubxn7, Faf1 is able to recover the recruitment of p97 and subsequently drive disassembly, thus acting in a redundant manner [89]. Of course, it is possible that both of these back-up mechanisms play a part.

#### 4.4.2. Processing Ubiquitylated CMG by p97

In order to understand how the length of the ubiquitin chain on Mcm7 affects the activity of the p97 complex, Fujisawa et al. (2022) established a reconstituted disassembly reaction, comprising CMG-Mcm7 conjugated with a single K48-linked chain of up to ~12 ubiquitins. When comparing the ability of yeast Cdc48^Ufd1/Npl4^ and human p97^UFD1/NPL4^ to disassemble this complex from beads, they found that the yeast complex was significantly more efficient than the human counterpart [88]. The support of p97^UFD1/NPL4^ activity by UBA-UBX cofactors UBXN7, FAF1 or FAF2, allowed for CMG disassembly to occur even when the CMG-Mcm7 substrate was modified with ubiquitin chains as short as 5 ubiquitins (Figure 3ii). This is equal to the earlier reported threshold of 5 ubiquitins needed for CMG unloading by yeast Cdc48, as well as the minimal chain length required for Cdc48 activity on artificial substrates [76,94]. Importantly, longer chains led to an increased efficiency of disassembly by p97 [76]. These findings thus underline and explain the importance of the additional cofactors in helping metazoan p97^UFD1/NPL4^ to efficiently bind and unfold ubiquitylated substrates.

Finally, several studies have enabled us to more precisely understand how p97 recognises and disassembles the terminated CMG. Firstly, it is evident that p97 complexed with cofactors primarily recognises the ubiquitin chains on Mcm7 rather than Mcm7 itself (Figure 3ii). This was proposed by Kochenova et al. (2022) and has been demonstrated by work from the Rapoport group, who used proteome-wide mass spec analyses and ubiquitin binding methods to conclude the same for all p97 substrates [89,95]. Further to this, cryo-EM structural analyses of Cdc48 revealed that, aided by Npl4′s MPN domain, the unfoldase initially binds to the N-terminus of the ubiquitin chain, rather than to the substrate [85]. This initial and reversible interaction causes the initiator ubiquitin molecule to unfold within the chain, which is subsequently pulled through the central pore of the ATPase channel, followed by the rest of the ubiquitin chain and the attached substrate [95] (Figure 3iii). Indeed, using a Cdc48-FtsH fusion protein, which is only able to degrade proteins once they have been translocated through the pore of the Cdc48 hexamer, Deegan et al. (2020) could confirm that budding yeast Cdc48^Ufd1/Npl4^ exclusively unfolds ubiquitylated Mcm7 from the CMG [76]. No other CMG subunits were found to be processed by Cdc48 in this manner, although Mcm4, which is positioned next to Mcm7, was found to be partially ubiquitylated and degraded.

Observations from reconstituted budding yeast reactions revealed that, while the CMG complex falls apart, the GINS complex remains intact and ubiquitylated Mcm7 remains bound to p97 [76]. Interestingly, Ji et al. (2022) observed that, following translocation, unfolded polyubiquitylated substrates are able to re-bind with Cdc48. This led the authors to speculate, given the observations made by Bodnar and Rapoport (2017), that DUBs are required to shorten the ubiquitin chain on the unfolded substrate in order to prevent another round of translocation [94,95]. Indeed, incubating the above reactions with the Cdc48-linked DUB Otu1, allowed release of the unfolded Mcm7 from Cdc48, supporting such a model [76,96,97]. In vivo studies using the *Xenopus* egg extract system, showed that incubation of the CMG complex with the NPE extract, led to CMG disassembly in an Lrr1- and p97-dependent manner, i.e., Flag-tagged MCM3 no longer interacted with the GINS complex. Intriguingly, however, they were able to detect interactions between MCM3 and MCM7 and speculated that the MCM2-7 complex had disassembled and reassembled in the timeframe of the experiment [42]. This might suggest that, following disassembly, components of the CMG are not degraded and are instead recycled, as purported by Fan et al. (2021); though it may simply demonstrate the fast nature of MCM2-7 formation in vivo [67]. As such, the fate of disassembled MCM7 is still an open question in the field.

### 4.5. Replisome Disassembly in Mitosis—The Back-Up Pathway

The deletion of Dia2 in budding yeast led to the retention of CMGs on chromatin until the following G1 phase [56]. In contrast to yeast, *C. elegans* early embryos unload CMGs in prophase of mitosis, when the canonical S-phase pathway is perturbed [64]. A similar back-up mitotic replisome disassembly pathway has now also been shown to exist in *Xenopus laevis*, mouse ES cells and human cells [66,68,98]. This pathway is capable of unloading replisomes at terminated or stalled replication forks and also relies on the same mechanisms/factors: ubiquitylation of MCM7 [98,99], CDC-48/p97/VCP unfoldase activity [64,68,98,99] and p97 cofactors NPL-4/UFD-1, as well as UBXN-3 in *C. elegans* [64], and UBXN7 and FAF1 in mouse ES cells [88] (Figure 5). Finally, while the evidence suggests that the SUMO protease ULP-4 supports this process in *C. elegans* [64], SUMOylation was found not to be required for the mitotic pathway functioning in *X. laevis* [98]. This could indicate species-specific differences in the regulation of this pathway or be a result of indirect mechanisms that are also species/system specific.

Given that MCM7 ubiquitylation and disassembly could be observed in mitosis, despite the inhibition of CUL2^LRR1^, this indicated that a secondary ubiquitin ligase was responsible for this role in mitosis [64]. Further investigations in *X. laevis*, *C. elegans* and mouse ES cells thus led to the identification of E3 TRAIP ubiquitin ligase as the specific factor driving replisome disassembly in mitosis [66,98,99,100] (Figure 5). Indeed, such studies in *X. laevis* demonstrated that TRAIP protects genome stability by not only promoting disassembly of replisomes at termination sites in mitosis, but also by supporting unloading at under-replicated regions of the genome and at stalled replication forks [98,99]. 

Finally, an analysis of MCM7 ubiquitylation in the mitotic pathway revealed that, in contrast to CUL2^LRR1^, TRAIP drives the formation of K6- and K63-linked polyubiquitin chains [98] (Figure 5). Although p97 is best known for processing substrates with K48-linked ubiquitin chains, alternative chain types do have the ability to recruit p97; indeed, recent work has revealed that p97 inhibition causes an accumulation of proteins comprising K48-K63-branched ubiquitin chains [101], demonstrating that there is still a great deal to be uncovered in this area.

#### Role of TRAIP during Incomplete Replication

DNA should be fully replicated during S-phase to allow for accurate chromosomes segregation and cell division in the subsequent mitosis. However, cells can enter mitosis with incompletely replicated DNA and stalled replication forks, and such a scenario is prevalent in cancer cells [102]. As mentioned above, TRAIP can ubiquitylate and drive the disassembly of any replisomes retained on DNA in mitosis [98,99], which is linked to mitotic replication fork breakage, processing, and mitotic DNA synthesis. Deng et al. noted that, in their experimental set up of “plasmid DNA replication in mitosis”, with mitotic CDK addition to *Xenopus* nucleoplasmic extract to resemble mitosis, mild replication stress led to the formation of aberrant DNA replication products (ARPs) [99]. The ARPs were the result of Polymerase theta-driven end-joining events between double strand breaks (DSBs) from different broken forks. In order to process stalled replication forks into DSBs the replisomes need to first be removed from DNA and this process was shown to be driven by TRAIP ubiquitin ligase [99]. The processing and rearrangements of such replication forks in mitosis could be either detrimental or beneficial for cell survival. Mitotic processing of a few under-replicated loci (e.g., late replicating common fragile sites) would allow for accurate chromosome segregation and maintain genome stability. However, similar processing of numerous stalled forks can lead to DNA fragmentation and cell death or cancer development. 

Under-replicated DNA can also be processed in early mitosis by a mitotic DNA repair synthesis (MiDAS) pathway [103]. In the presence of mild replication stress, cancer cells display DNA synthesis (EdU incorporation) in early mitosis, which is driven by the polymerases zeta and beta and depend on a prior processing of stalled DNA replication forks [104]. Work in early *C.elegans* embryos and in human cancer cells showed that TRAIP activity is needed for MiDAS. RNAi depletion of TRAIP led to the inhibition of MiDAS and generated anaphase and ultrafine anaphase DNA bridges. The cells also showed increased genomic instability in the form of micronuclei in the subsequent G1 phase of the cell cycle [100]. It seems, therefore, most likely that TRAIP-driven ubiquitylation and removal of the replisomes from stalled replication forks allows various nucleases and subsequent processes needed for MiDAS to access the DNA.

## 5. TRAIP Activity in S-Phase

Aside from TRAIP’s engagement in the removal of replisomes from DNA in mitosis, TRAIP also has the ability to disassemble CMGs in S-phase, initiating the repair of a specific subset of DNA inter-strand crosslinks (ICLs) [105]. ICLs are DNA lesions, which covalently link two DNA strands, creating a barrier to the passage of replication forks or transcription machinery. Collisions between converging replication forks and these ICLs have the potential to trigger two pathways of DNA repair, which involve either NEIL3 glycosylase or Fanconi anaemia (FA) proteins. NEIL3 can cleave a subset of ICLs (psoralen or AP-derived), while FA proteins unhook ICLs by creating incisions in the DNA strand around the ICL. Consequently, a DNA double-strand break (DSB) is created in one of the sister chromatids, which is then repaired through homologous recombination [106]. Investigations into ICL repair pathway choice in *X. laevis* egg extracts revealed the essential role of TRAIP E3 ubiquitin ligase. When two replication forks from neighbouring origins converge on a DNA crosslink and subsequently stall, CMG helicases undergo TRAIP-dependent ubiquitylation on the MCM2,3,4,6 and 7 subunits. These ubiquitin chains seem to be heterotypically linked and branched [105]. As described above, this is therefore very different to CUL2^LRR1^-driven MCM7 ubiquitylation at replication termination, comprising K48-linked ubiquitin chains, as well as TRAIP-driven MCM7 ubiquitylation in mitosis, which consists of K6- and K63-linked ubiquitin chains [55,98]. Further investigations led to the conclusion that NEIL3 glycosylase is recruited to the ICL by the short ubiquitin chains on CMG, enabling cleavage at the ICL. If NEIL3 is unable to unhook a crosslink, however, then the ubiquitin chains on CMG are extended and CMG is instead turned into a substrate for p97 unfoldase and subsequently unloaded from DNA. At this point, FANCI-FANCD2 proteins and endonucleases are recruited and ICL repair is processed through the FA pathway [105].

Another ubiquitin ligase, BRCA1 (Breast Cancer 1), was proposed to be essential for CMG unloading at ICL-stalled replication forks in *X. laevis* egg extracts [107]. However, it was later shown that the BRCA1 antibodies used to deplete BRCA1 from egg extracts in this set of experiments inadvertently also depleted the TRAIP ubiquitin ligase [105]. The CMG unloading defect observed in the egg extract depleted by these BRCA1 antibodies could be rescued by the addition of recombinant purified TRAIP [105]. Although BRCA1 does not, therefore, have a direct role in CMG unloading from stalled replication forks, it is likely to regulate this process through ubiquitylation of other replisome components, e.g., PCNA [108].

As well as being involved in ICL repair, TRAIP has also been found to help resolve DNA damage and replication stress in S-phase caused by a number of factors: UV radiation, which produces pyrimidine dimers and DSBs, hydroxyurea, which creates stalled replication forks, laser-induced DNA damage and endogenous damage caused by replication-transcription collisions [109,110,111,112]. In unperturbed conditions, TRAIP localises to the nucleoli and this localisation is dependent upon its catalytic activity [111]. Upon the creation of damage, however, TRAIP re-localises to the sites of DNA damage and promotes DNA damage signalling and repair [109,110,111]. Such localisation is dependent upon interactions with the sliding clamp loader PCNA though its PIP box, though it has also been shown to interact with other replisome factors, i.e., MCMs and the Replication Factor C (RFC) complex [110,111]. Currently the target(s) of TRAIP activity in these scenarios is yet to be clearly identified, but, at least in the removal of DNA-protein crosslinks (DPCs) from replicating chromatin, we know that TRAIP drives direct ubiquitylation of the DPC and proteolysis by the proteasome [113].

The crucial question that still needs to be answered in this field is how TRAIP activity is regulated. Given that TRAIP is capable of dismantling replisomes from chromatin at stalled or terminated replication forks, how does it distinguish those which are active at replication forks? As discussed above, CUL2^LRR1^ activity is regulated at the level of interaction with the substrate and can specifically only bind to terminated replisomes. TRAIP, however, seems to be present in the replication machinery throughout S-phase and mitosis. How TRAIP is regulated to perform its function only in strictly defined circumstances is therefore yet to be discovered.

## 6. Consequences of Defective Replisome Disassembly

Both *LRR1* and *TRAIP* are essential genes for cell proliferation and early embryo development in mice [114,115] (unpublished data Gambus lab). Moreover, homozygous mutations of the TRAIP ubiquitin ligase domain in humans can lead to microcephalic primordial dwarfism [109]. We cannot be sure, however, that these phenotypes are directly the result of defective replisome disassembly. Both TRAIP and LRR1 are likely to modify other cellular substrates as described in the sections above. As mentioned previously, faithful replisome disassembly throughout S-phase likely enables the recycling of rate-limiting replication factors. In support of this, the loss of LRR1 from human cells causes a reduction in the soluble levels of CDC45, POLE1 and TIMELESS and confers a significant reduction in global DNA synthesis [67]. Indeed, this latter phenotype has also been observed in cells lacking p97/CDC-48 activity [68,116] and it will be interesting to determine whether this reduction in DNA synthesis impacts late-firing replication origins in particular. 

Sequestering replisome components on the chromatin may also impact chromosomal segregation during mitosis and further phases in the cell cycle. The deletion of Dia2 in budding yeast led to the retention of CMGs on chromatin until the following G1 phase, which suggests that the cells were able to progress through mitosis. Nevertheless, this cell cycle progression was delayed and cells displayed chromosomal loss and chromosomal rearrangements, potentially as a result of unresolved DNA replication [56,117]. The loss of Dia2 from yeast also led to sensitivity to the DNA damaging agents MMS, HU and camptothecin, again suggesting problems with the DNA replication forks’ progression [117,118]. Further work is required to determine how these retained CMG helicases impact replication, transcription or other chromatin-related processes in the immediate and subsequent cell cycles in yeast.

In higher eukaryotes, TRAIP can remove any replisomes retained on chromatin past S-phase [98,100,104]. As described above, the inhibition of TRAIP activity leads to the inhibition of MiDAS, high levels of ultrafine and anaphase bridges and an increase in the occurrence of micronuclei [100]. All these are signs of genomic instability. TRAIP has also been reported to regulate mitotic progression; cells with downregulated TRAIP go through mitosis faster and with more chromosome segregation errors [119,120]. However, it is not known whether these mitotic progression errors are the consequence of retained replisomes on chromatin or another function of TRAIP in mitosis.

Finally, in *C. elegans* development, the loss of LRR-1 has been shown to cause cell cycle arrest, the loss of germ cells and sterility [121]. Authors have also observed increased levels of single-stranded DNA-binding RPA-1 nuclear foci, suggesting problems with DNA replication integrity. Low levels of LRR-1 depletion in *C. elegans* can be tolerated but the co-depletion of UBXN-3, which has been implicated in the mitotic pathway, leads to strong synthetic lethality, demonstrating that, upon the loss of S-phase disassembly pathway, the mitotic back-up pathway becomes essential for viability [64].

## 7. Future Directions

In the last ten years, we have made great strides towards understanding the processes involved in replication termination and replisome disassembly, both in S-phase and in mitosis, but there is still a great deal to be explored concerning the regulatory mechanisms. Aside from the essential question as to how the activity of TRAIP is regulated (explained above), several other interesting questions come to mind, making the next ten years of research into this area an exciting prospect:
1.Is there a role for deubiquitylating enzymes (DUBs) in preventing premature replisome disassembly?
The recent explosion of CRISPR screening is likely to direct us towards suppressors of problems generated by LRR1 or TRAIP downregulation and regulators or their activity. As the ubiquitylation field is also progressing very quickly, with numerous inhibitors for different ubiquitylation cascade enzymes and DUBs having been developed, there is much hope for using these and answering this question in the near future. 

2.What are the full consequences of inhibiting replisome disassembly in S-phase, making LRR1 an essential gene? Why is TRAIP unable to substitute for its loss?
The development of genome-wide analyses over the last few decades will be instrumental in answering questions about the efficiency of late-origin firing upon terminated replisomes retention, and in aiding our understanding of where and how the replication stress is generated upon LRR1 downregulation. We will also need all the help we can get from the recent surge in the development of structural biology approaches to understand the molecular aspects of TRAIP activity regulation.

3.What about dormant MCM2-7 helicases—how are they removed from DNA? While we know that they encircle DNA in a very stable manner and can be pushed ahead of replication forks, the mechanism of their unloading from chromatin is still a mystery.
We need a clever CRISPR screen that can measure retainment of the MCM2-7 on chromatin as a readout. 

4.Can the process of replisome disassembly be explored in the context of human health and disease? Replication initiation and elongation are frequently targeted in cancer therapy. Could we now also exploit our knowledge of replication termination to benefit human health?
The one factor that would most facilitate the delivery of translational potential of inhibition of replisome disassembly would be the development of small molecule inhibitors against TRAIP- and CUL2^LRR1^-driven ubiquitin ligase activity. Such inhibitors would allow for a quick and simple downregulation of their activities in any cell line, without the need for complex and time-consuming experimental setups.

## 8. Conclusions

While the previous 30 years of DNA replication research was dominated by uncovering the highly regulated process of replisome assembly, the last 10 years has seen a shift in focus towards understanding how these complex machineries are taken apart. Prior to 2014, we had very little understanding of what happens when two replication forks terminate. This review demonstrates how many developments have been made in the field over the last 10 years. It was a huge surprise, for example, to find that the terminated replisomes can pass each other upon fork convergence [26]. Ever since this finding, structural biology analyses of the organisation of the replisome and positioning of the leading and lagging strands, has provided us with a better understanding of how this process is coordinated. 

By far the best understood step in replication termination is the process of replisome disassembly, described first in 2014 [55,56]. We know that it is driven by ubiquitylation of MCM7 within the CMG helicase and we know which enzymes are involved in this process: the ubiquitin ligases and the p97 unfoldase together with its cofactors (UBXN7 and FAF1). While it was anticipated that a specific mechanism may exist to unload terminated replisomes, as active replisomes are otherwise protected from disassembly throughout S-phase, the level of regulation and sophistication of this mechanism has greatly surpassed our expectations.

## Figures and Tables

**Figure 1 biology-13-00233-f001:**
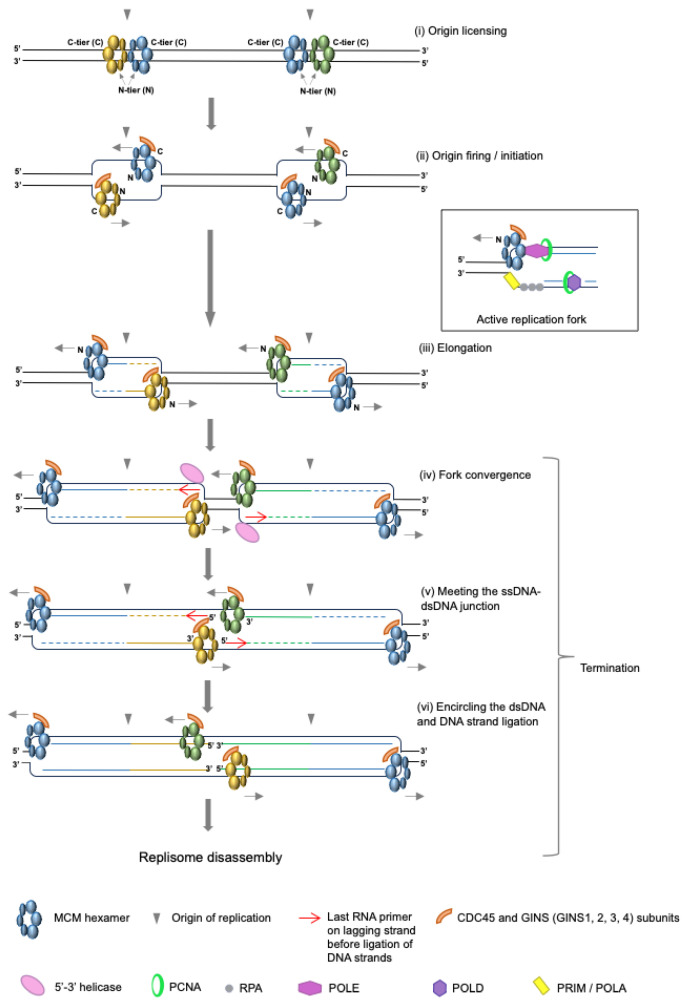
Steps of DNA replication. (**i**) Origins are selected along the chromosome and double MCM2-7 hexamers are loaded onto the DNA. C-tier: C-terminal tier of MCM2-7; N: N-terminal tier of MCM2-7. (**ii**) Following activation by binding of CDC45 and GINS (complex of SLD5, PSF1-3), CMG helicases, each built around a single MCM2-7 hexamer, bypass one-another and replication forks are created. Replication machinery at active replication forks is formed by hundreds of proteins. We depicted a few additional essential ones (e.g., DNA polymerases) in a side box. For the clarity of the figures, we retained only components of CMG helicase in further steps of the mechanism. (**iii**) CMG helicases progress through the chromatin, unwinding DNA in a 3′-5′ direction during elongation. (**iv**) Replication forks from neighbouring origins converge towards one-another as the DNA between them is unwound, with the aid of topoisomerases, fork rotation and 5′-3′ helicase enzymes. (**v**) The terminating CMG meets a ssDNA-dsDNA junction, comprising the RNA primer (red arrow) of the last, immature Okazaki fragment. (**vi**) The CMGs slide onto the dsDNA and translocate away, prior to disassembly, while the last stretches of DNA are synthesised and the final fragments are ligated together.

**Figure 2 biology-13-00233-f002:**
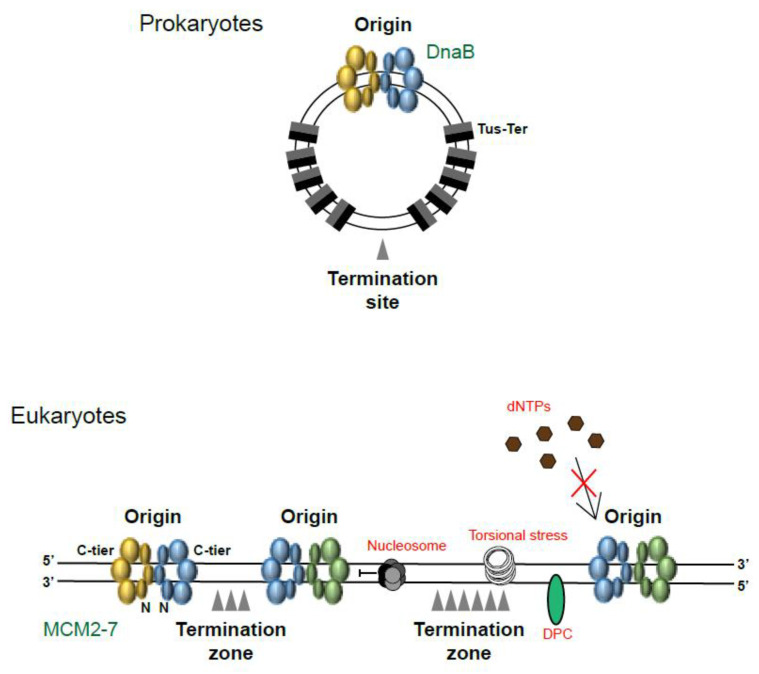
Initiation and termination zones. In prokaryotic circular mini-chromosomes, replication initiates from just one origin. The helicase (DnaB in bacteria and T-Ag in SV40 virus) first bypasses five ‘permissive’ (grey section head-on) Tus–Ter complexes, before reaching the termination site. From here, helicase progression is halted as it meets ‘non-permissive’ Tus–Ter complexes (black section head-on). In eukaryotic chromosomes, origins initiate at thousands of sights throughout the genome. Termination zones vary greatly as origins fire stochastically and progressing replication forks encounter many problems and obstacles along the way. C-tier: C-terminal tier of MCM2-7; N: N-terminal tier of MCM2-7; DPC: DNA-protein crosslink.

**Figure 3 biology-13-00233-f003:**
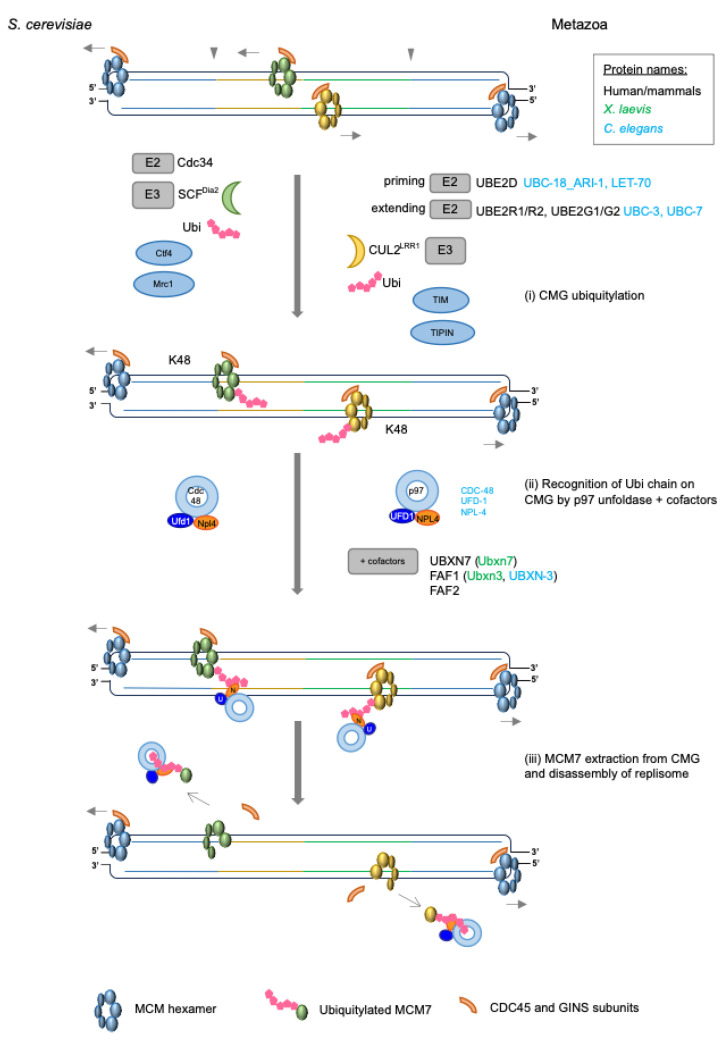
Mechanism of Cullin-driven replisome disassembly. (**i**) MCM7 is specifically polyubiquitylated with K48-linked polyubiquitin chains by Cullin ubiquitin ligase. In budding yeast, this ubiquitylation process engages SCF^Dia2^ E3 ligase, which cooperates with Cdc34 E2 conjugation enzyme, while recruitment relies upon Ctf4 and Mrc1. In metazoa, CUL2^LRR1^ E3 ligase, together with a range of priming and extending E2 conjugating enzymes, acts on the MCM7 substrate. In this case, TIMELESS and TIPIN replisome factors help in the recruitment of CUL2^LRR1^ to CMG and promote ubiquitylation events. (**ii**) Ubiquitylated CMGs are recognized by p97 unfoldase, which binds to its substrate by cooperating with a range of cofactors. (**iii**) The NPL4 subunit of p97 starts unfolding the ubiquitin chain on MCM7, which is then transferred through the channel of p97. The last step of this process involves extraction of MCM7 from the CMG helicase and removal/dismantling of the whole CMG from DNA.

**Figure 4 biology-13-00233-f004:**
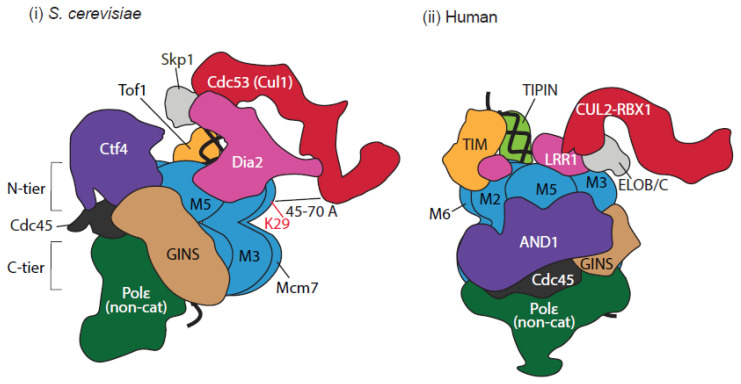
Mechanism of Cullin ubiquitin ligase interaction with terminated replisome. Schematic model representations depicting interactions between Cullin ubiquitin ligases and the budding yeast and human replisomes adapted from Jenkyn-Bedford et al.’s (2021) cryo-EM structures of complexes (reconstituted from purified proteins and assembled on a short fragment of dsDNA) [74]. Both Cdc53 and CUL2 display elongated configurations and conformational flexibility. (**i**) Budding yeast replisome interacting with Cdc53^Dia2^ (SCF^Dia2^); Dia2 interacts with Mcm3 and Mcm5 ZnF domains, with the Mcm7 N-terminus sandwiched between them. Due to the elongated forms and conformational flexibility of Cullin ligase, the Hrt1 (Rbx) component of the E3 ligase can be positioned at a close proximity to K29 of Mcm7, as indicated. (**ii**) Human replisome interacting with CUL2^LRR1^; the N-terminal pleckstrin homology domain of LRR1 interacts with the ZnF domains of MCM2 and MCM6, with dsDNA and with TIMELESS while the LRR region of LRR1 interacts with ZnF domains of MCM3 and MCM5 and the HMG-box of AND-1.

**Figure 5 biology-13-00233-f005:**
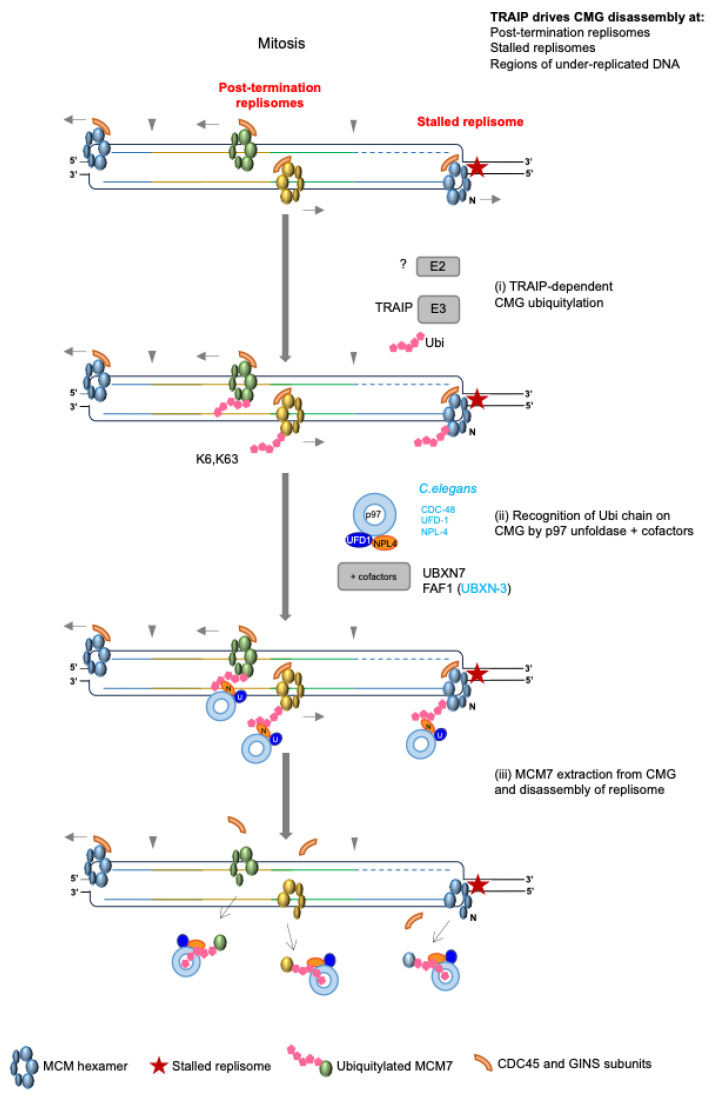
Mechanism of TRAIP-driven replisome disassembly. Replisomes retained on chromatin until mitosis in metazoa undergo TRAIP-dependent disassembly; these include post-termination replisomes, stalled replisomes or replisomes trapped in regions of under-replicated DNA. (**i**) Replisomes are ubiquitylated with K6- and K63-linked ubiquitin chains on the MCM7 subunit of CMG. This modification is dependent on the presence of TRAIP E3 ligase, although the E2 is currently unknown. (**ii**) Ubiquitylated CMGs become substrates for p97 unfoldase, which cooperates with a range of cofactors that aid in recognition and binding to ubiquitylated CMGs. (**iii**) Dismantling of the replisomes from chromatin starts when the NPL4 subunit of p97 unfolds the ubiquitin chain attached to the MCM7 subunit of CMG and then passes it through the p97 channel. The process ends with MCM7 being extracted from the replisome and disintegration/removal of the whole CMG from chromatin.

## Data Availability

No new data were created or analysed in this study. Data sharing is not applicable to this article.

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
