# Peer review of "A Decade of Discovery—Eukaryotic Replisome Disassembly at Replication Termination"

_biology, 2024, doi:10.3390/biology13040233_

Round 1

Reviewer 1 Report

Comments and Suggestions for Authors

The overall expression of the MS is like it is PhD preparation for the new topic of the future investigation. Such a work is usually necessary at the begging for the newcomer in the Lab. The review is interesting though could not be useful for the reader not aware of the topic. The reader should know the features of the proteins the authors are talking about. The advanced reader will get the idea of the future work direction. Still, the text should be corrected carefully. Right now, the references are of different types; all the references should be in one style. Too many sentences begin with word “interesting”. May be the statement is interesting for the authors, but better to use other expression.  The reviewer is not an English native speaker, but it was the surprise when word “however” stay in the middle of the sentence. Please, check whether it is correct. The pictures are good and informative, but their description is going on in the text far away of the picture itself. May be, it makes sense to mention that the picture described in details latter on. The format of references to the pictures is strange – why it is always in caps lock? Please, check whether it is correct according to the Journal’s rules. MS could be published while cleaned up and polished. Some bugs noticed are listed below.

Line 24 -  MCM2-7 helicase complex – abbreviation should be defined. It is used in figure legend under Fig.1, i.e. quite soon

Line 30 -   (FIGURE 1) – why in CAPS LOCK?

33 -   (McGuffee, Smith et al., 2013)- what a strange format of reference?

Fig 1 legend:  CDC45 and GINS – while CDC45 is more or less clear, GINS should be defined

49, 103, 131 - (FIGURE 2).- again  CAPS LOCK

51 - Tus-Ter protein-DNA complexes – should be defined, i.e. what is it? Otherwise it is not clear what is on Fig. 2.

55 - (Berghuis, Raducanu et al., 2018) and (Rudolph, Corocher et al.) – 1st reference is in the form throughout the whole MS and it looks strange. It is up to the Editor to decide whether they are correct or not. The 2nd one is definitely wrong

61 -  bacterial chromosome however, with new initiation – looks like incorrect expression

62 - (Rudolph et al.) – reference

Figure 2 legend is not complete. What is DPC? What is C-tier? DPC definition found at line 130. It should be in the figure legend. C-terminal tier is found at 211 line.

165 - Sundin & Varshavsky, 1980, Sundin & Varshavsky, 1981-  Should be Sundin & Varshavsky, 1980,1981. Why sometimes “,” (comma), sometimes “&”? Should be one style.

174, 177 -  at the 1st time, please, note that Topoisomerase II enzymes will be TopII enzymes below

195 - FIGURE 1iv - why in CAPS LOCK? Comma, please – 1, iv

207 - FIGURE 1iv – vi - Comma, please

213 - N-terminal tier is absent at Fig.1

251- RPA dissociation- what is RPA? Definition, please.

352- cryo-EM- is it cryo-electron microscopy?

356- (discussed further below)- may be it’ll be useful to insert more such a notes, especially for figures

385-388 – font different

389 – is anybody know what precise information produced by the Brown and Walter labs?

441- please, check whether CDK was defined

550 – is   Faf1 here the same item as  FAF1 at, say, line 515 or 562?

610 -  is CDC45    identical to Cdc45 at, say, line 407?

626 – references - Priego Moreno S, and  Moreno SP – is it the same person or not?

667 - a great deal to be uncovered in this area – what means the reference to this sentence?

696 - pytimidine dimers – check the term

708 – DPC definition was at the line 130, though could be forgotten.

Author Response

We would like to thank the reviewer for generally positive view of our review and recognition that it will be “useful”. Below we address the suggestions of the reviewer:

  1. the references are of different types; all the references should be in one style.

All references are now in the ACS format as requested by the editor.

  1. Too many sentences begin with word “interesting”. May be the statement is interesting for the authors, but better to use other expression. 

This has now been changed and “interesting” has been removed throughout the manuscript.

  1. The reviewer is not an English native speaker, but it was the surprise when word “however” stay in the middle of the sentence. Please, check whether it is correct.

When using the word 'however' as an adverb, to mean 'despite this', it can be placed in the middle of a sentence, as described in the online Cambridge dictionary. Generally, the positioning of the word in a sentence is up to the personal preference of the writer.

  1. The pictures are good and informative, but their description is going on in the text far away of the picture itself. May be, it makes sense to mention that the picture described in details latter on.

The positioning of figures within the manuscript is provided by the editorial office. We have made sure, however, that we cite the figures more often within the text, so that the reader is directed towards them sooner.

  1. The format of references to the pictures is strange – why it is always in caps lock?

All references to figures now changed to “Figure…”

  1. Line 24 - MCM2-7 helicase complex – abbreviation should be defined. It is used in figure legend under Fig.1, i.e. quite soon 

Now added “MCM2-7 (minichromosome maintenance 2-7) helicase” into section 1.

  1. Line 30 -  (FIGURE 1) – why in CAPS LOCK?

All references to figures now changed to “Figure…”

  1. 33 -   (McGuffee, Smith et al., 2013)- what a strange format of reference?

All references are now in the ACS format as requested by the editor.

  1. Fig 1 legend:  CDC45 and GINS – while CDC45 is more or less clear, GINS should be defined

GINS is now defined as: complex of SLD5, PSF1-3 in the figure legend.

  1. 49, 103, 131 - (FIGURE 2).- again  CAPS LOCK

All references to figures now changed to “Figure…”

  1. 51 - Tus-Ter protein-DNA complexes – should be defined, i.e. what is it? Otherwise it is not clear what is on Fig. 2.

Now added: “The Tus-Ter barriers consist of high affinity binding of Tus protein to specific DNA sequences called Ter. Such barriers can be orientated in a permissive orientation, permitting the bypass of the DnaB replicative helicase, or in a non-permissive orientation, which blocks the progression of DnaB. Following initiation, each fork bypasses five ‘permissive’ Tus-Ter protein-DNA complexes until they reach the stopping region. From this point onwards, progression is halted as the forks encounter the ‘non-permissive’ Tus-Ter complexes.”

  1. 55 - (Berghuis, Raducanu et al., 2018) and (Rudolph, Corocher et al.) – 1streference is in the form throughout the whole MS and it looks strange. It is up to the Editor to decide whether they are correct or not. The 2nd one is definitely wrong

All references are now in the ACS format as requested by the editor.

  1. 61 -  bacterial chromosome however, with new initiation – looks like incorrect expression

Now written as: “..bacterial chromosome, with new initiation..”

  1. 62 - (Rudolph et al.) – reference

All references are now in the ACS format as requested by the editor.

  1. Figure 2 legend is not complete. What is DPC? What is C-tier? DPC definition found at line 130. It should be in the figure legend. C-terminal tier is found at 211 line.

Now added: “C-tier: C-terminal tier of MCM2-7; N: N-terminal tier of MCM2-7; DPC: DNA-protein crosslink.”

  1. 165 - Sundin & Varshavsky, 1980, Sundin & Varshavsky, 1981-  Should be Sundin & Varshavsky, 1980,1981. Why sometimes “,” (comma), sometimes “&”? Should be one style.

All references are now in the ACS format as requested by the editor.

  1. 174, 177 -  at the 1sttime, please, note that Topoisomerase II enzymes will be TopII enzymes below 

Now written as: “Topoisomerase II (TopII)”

  1. 195 - FIGURE 1iv - why in CAPS LOCK? Comma, please – 1, iv

All references to figures now changed to “Figure X, xx”

  1. 207 - FIGURE 1iv – vi - Comma, please

All references to figures now changed to “Figure X, xx”

  1. 213 - N-terminal tier is absent at Fig. 1

“C-tier: C-terminal tier of MCM2-7; N: N-terminal tier of MCM2-7” added to Figure 1 legend.

  1. 251- RPA dissociation- what is RPA? Definition, please.

Now written as: “This ligation was found to occur in the same timeframe as RPA (Replication Protein A) dissociation from the DNA. As RPA binds and protects ssDNA from breakage and degradation, this dissociation pattern suggests that RPA coats and protects sequences of ssDNA within the termination zone [52,53].

  1. 352- cryo-EM- is it cryo-electron microscopy?

Now written as: “…applied cryo-electron microscopy (cryo-EM)..”

  1. 356- (discussed further below)- may be it’ll be useful to insert more such a notes, especially for figures.

We inserted more directions to figures within the text.

  1. 385-388 – font different

Changed

  1. 389 – is anybody know what precise information produced by the Brown and Walter labs?

Now written as: “..produced by the Brown and Walter labs; they co-expressed of all 5 subunits in the baculovirus system and cryo-EM analysis revealed that Cul2Lrr1 possesses a long, open architecture.”

  1. 441- please, check whether CDK was defined

Now written as: “CDK (Cyclin-Dependent Kinase)” in Section 4.3.

  1. 550 – is   Faf1 here the same item as FAF1 at, say, line 515 or 562?

It is the same protein. We switch between formats, depending on whether we’re talking about yeast or Xenopus proteins (Faf1) or human proteins (FAF1).

  1. 610 -  is CDC45 identical to Cdc45 at, say, line 407?

It is the same protein. We switch between formats, depending on whether we’re talking about yeast or Xenopus proteins (Cdc45) or human proteins (CDC45).

  1. 626 – references - Priego Moreno S, and  Moreno SP – is it the same person or not? 

Yes, the same person. Reference manager cites them differently. This is not a problem now as all references are now in the ACS format as requested by the editor.

  1. 667 - a great deal to be uncovered in this area – what means the reference to this sentence?

Now written as: “..p97 inhibition causes an accumulation of proteins comprising K48-K63-branched ubiquitin chains [102], demonstrating that there is still a great deal to be uncovered in this area.” At the end of section 4.5.

  1. 696 - pytimidine dimers – check the term

Changed to “pyrimidine dimers”.

  1. 708 – DPC definition was at the line 130, though could be forgotten.

Definition for DPC now in three places throughout the review.

Reviewer 2 Report

Comments and Suggestions for Authors

The review is very comprehensive and detailed. It does touch upon the key stages associated with replication termination and CMG disassembly. However, the authors should also include other key factors like PCNA and polymerases especially in the Figure 1. Similarly, the readers can also benefit if the authors include a short paragraph on the role of BRCA1 in CMG disassembly similar to the role of TRAIP during incomplete DNA replication. 

Author Response

We would like to thank reviewer for the comment: “The review is very comprehensive and detailed.” Thank you for the constructive suggestions.

  1. However, the authors should also include other key factors like PCNA and polymerases especially in the Figure 1.

We added a side box with a model of the replisome with polymerases and PCNA.

We also added the explanation in the figure legend:

“Replication machinery at active replication forks is formed by hundreds of proteins. We depicted a few additional essential ones (e.g. DNA polymerases) in a side box. For the clarity of the figures, we retained only components of CMG helicase in further steps of the mechanism.”

  1. Similarly, the readers can also benefit if the authors include a short paragraph on the role of BRCA1 in CMG disassembly similar to the role of TRAIP during incomplete DNA replication. 

Both of these sections have now been added to the review – thank you for the suggestion.

Reviewer 3 Report

Comments and Suggestions for Authors

In this manuscript, authors presented a comprehensive review of the mechanisms involved in the termination of DNA replication in various model systems. The authors first introduced the process of DNA replication, from initiation through elongation to termination, and then discussed each step in the replication termination process in detail. At the end, they proposed several questions regarding the future directions for replication termination and replisome disassembly. Below are some suggestions for enhancement.

1. Although the manuscript is rich in detail and well structured, it could benefit from a brief introduction to key concepts, such as CMG (Cdc45-MCM-GINS) complex, at the beginning and a summary at the end, making the manuscript accessible to a broader audience, including those who are not in relevant fields.

2. Given the critical role of CMG complex in replication, authors should consider elucidating the regulatory mechanisms underlying the disassembly process of CMG complex. Additionally, a section focusing on how incorrect CMG disassembly contributes to replication stress and impacts genome stability can give the readers more insight into the importance of proper CMG complex disassembly.

3. The questions raised at the end of the manuscript are insightful. However, it would be beneficial to expand this section to include potential experimental approaches or technologies that could be used to address these questions. For example, discussing the role of CRISPR-Cas9 based genomic manipulation could inspire future work for researchers.

Overall, this manuscript makes a valuable contribution to the field of DNA replication, provides a thorough and insightful review of current knowledge and identifies exciting directions for future research.

Author Response

We would like to thank the reviewer for the statement that “the manuscript is rich in detail and well structured”. To address the reviewer’s comments:

  1. The manuscript could benefit from a brief introduction to key concepts, such as CMG (Cdc45-MCM-GINS) complex, at the beginning and a summary at the end, making the manuscript accessible to a broader audience, including those who are not in relevant fields. 

We included more explanation in the first paragraph of section 1. We aimed this review as one in a series within the themed issue, but we appreciate that additional explanations benefit the understanding.

Also, a conclusion has been written for the paper, as also requested also by the editor.

  1. Given the critical role of CMG complex in replication, authors should consider elucidating the regulatory mechanisms underlying the disassembly process of CMG complex.Additionally, a section focusing on how incorrect CMG disassembly contributes to replication stress and impacts genome stability can give the readers more insight into the importance of proper CMG complex disassembly. 

We feel we have discussed at length the regulatory mechanisms underlying the disassembly process. We have now separated out into an additional section the information discussing the consequences of defective replisome disassembly – replication stress and genome instability. We extended this discussion too.

  1. The questions raised at the end of the manuscript are insightful. However, it would be beneficial to expand this section to include potential experimental approaches or technologies that could be used to address these questions. For example, discussing the role of CRISPR-Cas9 based genomic manipulation could inspire future work for researchers.

We have now included such discussions for each question proposed.